# Identifying barriers and facilitators in HIV-indicator reporting for different health facility performances: A qualitative case study

**Milka B. Gesicho** [1,2]☯*, **Ankica Babic** [1,3]☯

**1** Department of Information Science and Media Studies, University of Bergen, Bergen, Norway, **2** Institute of Biomedical Informatics, Moi University, Kesses, Kenya, **3** Department of Biomedical Engineering, Linköping University, Linköping, Sweden

☯ These authors contributed equally to this work.
* milcagesicho@gmail.com

**Data Availability Statement:** All relevant data are within the paper.

**Funding:** This work was supported in part by the NORHED program (Norad: Project QZA-0484). link: (https://www.norad.no/en/front/funding/norhed/

## Abstract

Identifying barriers and facilitators in HIV-indicator reporting contributes to strengthening HIV monitoring and evaluation efforts by acknowledging contributors to success, as well as identifying weaknesses within the system that require improvement. Nonetheless, there is paucity in identifying and comparing barriers and facilitators in HIV-indicator data reporting among facilities that perform well and those that perform poorly at meeting reporting completeness and timeliness requirements. Therefore, this study aims to use a qualitative approach in identifying and comparing the current state of barriers and facilitators in routine reporting of HIV-indicators by facilities performing well, and those performing poorly in meeting facility reporting completeness and timeliness requirements to District Health Information Software2 (DHIS2). A multiple qualitative case study design was employed. The criteria for case selection was based on performance in HIV-indicator facility reporting completeness and timeliness. Areas of interest revolved around reporting procedures, organizational, behavioral, and technical factors. Purposive sampling was used to identify key informants in the study. Data was collected using semi-structured in-depth interviews with 13 participants, and included archival records on facility reporting performance, looking into documentation, and informal direct observation at 13 facilities in Kenya. Findings revealed that facilitators and barriers in reporting emerged from the following factors: interrelationship between workload, teamwork and skilled personnel, role of an EMRs system in reporting, time constraints, availability and access-rights to DHIS2, complexity of reports, staff rotation, availability of trainings and mentorship, motivation, availability of standard operating procedures and resources. There was less variation in barriers and facilitators faced by facilities performing well and those performing poorly. Continuous evaluations have been advocated within health information systems literature. Therefore, continuous qualitative assessments are also necessary in order to determine improvements and recurring of similar issues. These assessments have also complemented other quantitative analyses related to this study.

projects/health-informatics-training-and-research-in-east-africa-for-improved-health-care-hi-train/)
The funders had no role in study design, data collection and analysis, decision to publish, or preparation of the manuscript.

**Competing interests:** The authors have declared that no competing interests exist.

## Introduction

In a bid to eradicate the HIV epidemic, enormous strides have been made toward achieving the UNAIDS "90 90 90" ambitious target [1]. The goal of these targets was that by 2020, "90% of all people living with HIV will know their HIV status; 90% of all people with diagnosed HIV infection will receive sustained antiretroviral therapy; and 90% of all people receiving antiretroviral therapy will have viral suppression" in order to end the epidemic by 2030 [1]. Therefore, tracking the national response of HIV is salient in determining progress as well as recognizing whether efforts to scale up HIV services are of value. As such, Monitoring and Evaluation (M&E) systems, which are regarded as the cornerstone of HIV services, have been established in low and middle-income countries (LMICs) to provide high quality strategic information for decision-making [2,3].

To measure HIV program effectiveness and patient outcomes, Ministries of Health (MoH), as well as international donor organizations, such as Presidential Emergency Plan for AIDs Relief (PEPFAR) require the various health facilities to report several aggregated indicators as part of M&E program [4]. The scale up of HIV services in LMICs has resulted to strengthening of Health Management Information Systems (HMIS) to facilitate with collection, management, and availability of timely, complete and accurate data. Therefore, HMISs such as the District Health Information Software Version 2 (DHIS2) have been implemented in over 70 countries to promote availability of routine aggregated indicator data within health care [5]. As such, routine reporting of HIV-indicators in many LMICs is performed using the District Health Information Software Version 2 (DHIS2) [6].

Therefore, aggregate indicator data from various HIV services are collected using paper-based summary forms at the facility level, which are then entered into the DHIS2. These data are required to be submitted monthly to the national level by health facilities within stipulated timelines. As such, the HIV-indicator data are used for M&E by converting the raw data collected, to information for utilization in decision-making. Nevertheless, M&E systems in LMICs experience significant challenges in completeness, timeliness and accuracy in reporting data [2,7]. Some of these challenges are often brought about by various issues revolving around organizational, technical and behavioral factors [8].

Moreover, identifying barriers and facilitators in HIV reporting contributes to strengthening HIV M&E efforts by acknowledging contributors to success, and identifying weaknesses within the system that require improvement [3,9–12]. As such, various assessments have been conducted in a bid to improve performance of facilities at meeting data quality reporting requirements such as accuracy, completeness and timeliness, which are essential for M&E [9,13–15]. Nonetheless, qualitative assessments that follow these evaluations are meagre. This will facilitate understanding similarities or differences across various issues within facilities with varying performances, given that some facilities perform better in meeting data quality reporting requirements than others. As such, there is paucity in identifying and comparing barriers and facilitators in HIV-indicator reporting among facilities that perform well and those that perform poorly at meeting reporting requirements such as completeness and timeliness.

This study aimed at conducting qualitative case study using Kenya as an example, to identify and compare the current state of barriers and facilitators in routine reporting of HIV-indicators, based on facility performance. This was conducted among facilities performing well, and those performing poorly in meeting DHIS2 reporting requirements (facility reporting completeness and timeliness). The findings of this study aimed at contributing towards strengthening HIV-M&E efforts, which are of interest to various stakeholders including ministries of health.

## Method

### Study setting

This study was conducted in Kenya's capital Nairobi. Kenya is a sub-Saharan country in East Africa with 47 administrative counties, and it's health system is categorized in six levels, which include (i) community services, (ii) dispensary/clinics, (iii) health centers, (iv) sub-county hospitals, (v) county referral hospitals, and (vi) national referral hospitals [16,17].

### Study design

A qualitative case study approach was used. Miles and Huberman define a case as "a phenomenon of some sort occurring in a bounded context" [18]. Creswell on the other hand defines a case as "a program, organization, event, activity, process, or one or more individuals" [19]. As such, the "case" in this study is defined as a health care facility offering HIV services. The cases in this study were bounded by context, which includes only HIV healthcare facilities that meet the following criteria (i) located in Nairobi (ii) either use EMRs system or paper in reporting, (iii) reporting performance (facility reporting completeness and timeliness).

### Data collection

Purposive sampling was used to identify the cases (health facilities) from constituencies in Nairobi [20]. In addition, the cases were also drawn from level two and level three of the health system, which comprises of clinics and health centers. Hence, the type of purposive sampling used was stratified purposeful sampling, whereby health facilities from the two levels were selected based on reporting performance [21].

Thus, these cases (health facilities) were selected based on performance in facility reporting completeness and timeliness of Care and Treatment reports for the years 2017 and 2018 with more details outlined in Gesicho et al. [21,22]. Facility reporting completeness was defined as the percentage of actual reports submitted to DHIS2 against the expected reports, whereas facility reporting timeliness was defined as the percentage of actual reports submitted to DHIS2 on time, against the expected reports.

In Kenya, HIV-indictor reports are submitted in DHIS2 on a monthly basis by facilities offering HIV services using the MOH-mandated form called "*MOH731- Comprehensive HIV/AIDS Facility Reporting Form*" (MOH-731). Care and Treatment is among the most reported and salient HIV services offered by health facilities in Kenya. The criteria for selection was based on HIV reporting performance by facilities, which was categorized as best performers, average performers, poor performers and outlier performers. This grouping was based on a cluster analysis conducted in order to evaluate the reporting performance by facilities using completeness and timeliness as performance indicators as outlined in Gesicho et al. [21,22]. In this study, reporting performance was categorized into two main groups, facilities performing well (best performers = 3 facilities, average performers = 3 facilities) and facilities performing poorly (outlier performers = 4 facilities and poor performers = 2 facilities). One health facility where a sub-county office is located was also included to provide more information on reporting by facilities as all facilities are required to submit paper-based reports to their respective sub-county office.

Purposive sampling was also used to identify key informants in the study in order to conduct in-depth interviews [20]. Therefore, the key informants who in this study are the units of analysis included personnel in charge of reporting as they serve as the focal point around which all reporting activities take place.

Data was collected using semi-structured in-depth interviews with 13 participants, in 13 health facilities, with one facility visit being specifically to the sub-county office. These were

drawn from six of the 17 constituencies in Nairobi, which include Kasarani, Embakasi North, Embakasi South, Embakasi East, Embakasi Central, Kamukunji, Dagoretti South, and Dagoretti North. Archival records on facility reporting performance, documentation and informal direct observation were also used to collect the data. These sources of evidence aimed to ensure improved credibility through data triangulation. Archival records such as retrospective quantitative data on facility reporting timeliness and completeness were retrieved from DHIS2 in order to identify the reporting performance of a facility. Documentation such as hard copy standard operating procedures, and data quality assessments (DQA) reports were sought and perused. Informal observations were also carried out during the assessments, which involved observing the documents put on office walls, office space, interaction with colleagues and environment where the facility was located.

The in-depth interviews were conducted using an interview guide. The interview guide explored the technical, behavioral, and organizational factors, which facilitated or hindered the reporting process in the health facilities based on participants' experiences and perceptions. Interviews took place in enclosed workstations of the participants and lasted approximately between one to one and a half hours. Data saturation was achieved in the sampled cases given that no new data emerged during the final interviews. All data were collected between September 2019 and November 2019.

### Interview guide

The interview guide used in this qualitative assessment was based on the conceptual framework for performance of the routine information system management (PRISM), which was developed to strengthen routine health information system (RHIS) performance management [8]. A routine health information system (RHIS) is comprised of inputs (RHIS determinants), processes (RHIS process) and outputs (improved RHIS performance), which are components of a routine health information system [8]. We explored the RHIS determinants of performance and RHIS process specified within the PRISM framework. The RHIS determinants include technical (complexity of reporting form, procedures), behavioral (competence, motivation) and organizational determinants (availability of resources, training). The RHIS process elements explored include: data collection process, data processing, data transmission and data quality checking and feedback mechanisms. This study aimed to identify and compare the barriers and facilitators linking to RHIS determinants and process, among facilities in the four performance cluster categories (best performer, average performers, poor performers, and outlier performers). For detailed information on the interview guide (See S1 Appendix: Interview guide).

### Ethical considerations

Ethical approval was obtained from the Institutional Research and Ethics Committee in Moi University-No.0003362. Other approvals were obtained from the Ministry of Health, Nairobi County and from affiliated constituencies where the facilities were sampled from. Research license was obtained from the National Commission for Science, Technology & Innovation in Kenya. Privacy and confidentiality were ensured by not revealing the identities of the participants nor the facilities that took part in this study. Written informed consent was obtained from all participants who were interviewed.

### Data analysis

Data from interviews and data sources were analyzed together. This followed the analysis framework developed by Morse, which outlines four key cognitive processes used in

developing theory from data [23]. These processes include comprehension, synthesis, theorizing and recontextualizing. The process of comprehending begins during data collection. Comprehension also involves coding, which enables sorting data, and uncovering underlying meanings in the text [23]. The synthesis process involves aggregating several stories or cases to describe typical, composite patterns [23]. Content analysis was used in the comprehension and synthesis processes. A provisional 'start list' of codes was created based on the research questions in order to make the coding process manageable. Theorizing involves a systematic process that entails finding alternative explanations until an explanation that best fits the data is sought [23]. Within-case and cross-case comparisons were used in the theorizing process by using cross case displays presented by Miles and Huberman [18]. Only analyzed cross-case data were presented in order to ensure that the confidentiality of sites, which may be identifiable from the within-case analysis. Re-contextualization involved comparing the findings with previous research in order to enhance trustworthiness. QSR NVivo was chosen as the Computer-Assisted Qualitative Analysis Software (CAQDAS), were all data was managed [24].

## Trustworthiness

Various approaches were used in order to ensure trustworthiness of the study. To ensure credibility of the study, prolonged engagement with participants was conducted long enough to gain trust and establish rapport [25]. Triangulation of data sources was conducted using aforementioned multiple sources. Peer debriefing was also carried out in sessions after conducting two to three interviews and during analysis [25]. To ensure dependability and confirmability, an audit trail using QSR NVivo (Version 12), was used to manage and store data. Reflexivity was achieved on the basis that the researcher was a stranger in the facility settings hence had no established familiarity with participants prior to the study, which may be a threat to validity (bias) [25]. Transferability was attained through provision of thick descriptions hence enabling applicability of findings in other similar settings using DHIS2 to submit monthly reports.

## Result

A summary of the findings is presented in Table 1.

It is worth noting that the key respondents interviewed identified their positions as either data officers or M&E assistants but performed the same tasks. Hence, the position title depended on the term used by facility or supporting partner in charge of a facility. Thus, the term 'data officers' is used as a general term in this study to refer to the people mandated with data reporting process in health facilities. Detailed findings revolving around technical, organizational, and behavioral factors are outlined below.

### Interrelation between workload, teamwork, and capacity in reporting

Emerging interrelated factors that influenced the reporting process in the various facilities include workload, teamwork, and human resource capacity. As such, the presence or absence of a combination of these factors either positively or negatively influenced the reporting process. Workload referred to the amount of work present in each health facility and this varied from facility to facility with some having more workload than others. Teamwork in this case means that the people involved in reporting assist each other in collection, aggregation, and verification of data. For instance, in some cases nurses at the various service points are required to update correct aggregate numbers for indicators and submit them on time during the reporting period to the data officer for verification. An informal observation made when conducting interviews was the interaction between the nurses, clinicians and the data officer, which revealed a sense of putting in effort to provide the data required by the data officers. For

**Table 1. Summary of findings on barriers and facilitators in HIV-indicator reporting.**

| Factors | Category | Summary of key findings |
|---|---|---|
| 1. Interrelationship between workload, teamwork, and capacity in reporting | Organizational | Presence of teamwork as well as sufficient number of skilled personnel in facilities with a lot of workload emerged as a facilitator in the reporting process regardless of facility performance group. |
| 2. Role of an EMRs system in easing the reporting process | Reporting process, Technical | Presence of an EMRs system facilitated the ease of data collection in the reporting process. Nonetheless, it did not equate to good performance as two of the facilities in the best performance group were not using an EMRs system whereas all facilities in the outlier performance group had an EMRs system. |
| 3. Reporting timeframe and adherence to reporting deadlines | Reporting process | Reporting days falling on weekends emerged as a major time constraint in the reporting process among most of the facilities visited. Facilities in the various performance groups echoed the same issue. |
| 4. Access rights and availability of national aggregate reporting system | Technical | Lack of access rights to DHIS2 by facilities was a contributor to late submission of reports. This is especially when reports submitted by hand to the sub-county are entered late in DHIS2. Lack of availability of DHIS2 due to issues such as system down times and lagging internet contributed to slowing down the reporting process. |
| 5. Complexity of reports, staff rotations, and role of mentorship in reporting | Technical | Documentation errors were among the main issues resulting from these factors regardless of facility performance group. |
| 6. Fit between individual, task and technology in reporting | Technical | Fit between individual task and technology is a facilitator in reporting. Facility that lacked fit between individual and task reported lack of motivation. |
| 7. Motivation and awareness of reporting performance | Behavioral | Data officers and M&E assistants interviewed used ad hoc approaches to determine their individual performance in submitting reports. For instance, some of them mentioned that once they have submitted a report on time and with no questioning of the data, then they have performed well. Nonetheless, facilities generally depended on key administrators such as in-charges to provide feedback on performance. Good feedback was a motivating factor. |
| 8. Availability of Standard Operating Procedures, Training, and Supervision | Organizational | Regardless of reporting performance, facilities funded by partners had SOPs and on job trainings whereas those not funded did not. Supportive supervisions were also reported to be present but not frequent. |

instance, in some facilities, an observation was made on nurses bringing updates on aggregated data written on paper to the data officer On enquiry, the interviewee responded that regular updates promoted data quality as it enabled accountability when performing weekly and monthly verification of patient numbers per indicator.

If this teamwork is not present, the data officer may have a difficult time in collecting the aggregated indicator data. On the other hand, human resource capacity referred to sufficient number of skilled personnel involved in the reporting process. Hence, the interrelationship comes about in the sense that more workload required skilled human resource as well as team-work in order to get the job done.

As such, respondents attributed time-consuming data collection process to more workload. The amount of workload was determined by factors such as services offered as well as the number of patients visiting the facilities. Health facilities with well-established Comprehensive Care Centers (CCCs) offered more HIV services and therefore received more patients. Hence, such facilities required to use more registers due to many service points as well as patient files. As such, human resource capacity as well as teamwork is important as stated by some of the respondents:

> "You know in other facilities, you find the clinician is one, he is the one who is the pmtct, he is the one seeing all other CCC patients, sometimes they have only one or two counselors depending on the workload. Workload in facility X CCC alone has 3500 patients, we have 3 clinicians, two nurses, 8 counselors. So those are so many registers."- **Data officer facility K (outlier performer)**

> "Basically I should be doing the reports but since the testing points are so many, they do compile their report for the partner and give me a copy"- **Data officer facility C (best performer)**

As such, having to deal with many registers can also affect the quality of data since it is time consuming for the data officer to perform assessments given that there was only one data officer in most facilities visited. As stated by one of the respondents:

*"Ideally, I am supposed to go register by register counting, that is a quality assessment. Am supposed to be counting. At times, I may not be having that time sincerely. But the supervisor is also thorough."*—**Data officer facility A (best performer)**

Teamwork then plays a big role in facilities that have a lot of workload. For teamwork to be effective, there needs to be trained personnel involved, who have a good knowledge and understand the indicators. This limits the documentation issues hence making the process faster as there will be minimal corrections in the data as reported indictors will be tallying. In cases where teamwork is present, but the personnel involved in reporting process do not have a good knowledge and understanding of indicator data, issues of documentation frequently arise. Moreover, in some cases the data officer has to do the collection and verification of indicator data as there is no one to provide a summarized report, which then consumes time taken to collect data. This then slows down the reporting process as data needs to be verified and amended.

Moreover, respondents in facilities that used an EMRs system for retrospective data entry cited backlogs due to accumulation of patient files, which need to be input to the system before the reporting period. The data in the systems need to be up to date in order to generate reports and queries needed for populating the MOH-731. In most cases, there was only one person who was assigned with the retrospective data entry to the EMRs system, which then contributed to backlog when they were absent from duty. Some of the data officers interviewed mentioned that they felt overwhelmed especially during the reporting period. As stated by one of the participants:

*"Like this week, I will get a huge back log because I have to generate the reports so by the time I am generating the reports I will not do data entry. You know I am the same person doing data entry of the daily activities and then I am the one who generates the reports. So, the reporting week I am always getting overwhelmed"*–**Data officer facility J (average performer)**

Hence, teamwork, as well as sufficient number of skilled human resource is salient especially in facilities with a lot of workload as this can either slow down or speed up the reporting process.

### Role of an EMRs system in easing the reporting process

Some of the facilities visited, had EMRs systems implemented as point of care (one facility) or retrospective data entry (four facilities). In other facilities, the EMRs systems were non-functional meaning that they were present but not being utilized in reporting or for other tasks in the facility (three facilities). In the facility where EMRs system was used as point of care, data entry was done by clinicians. Nevertheless, the data officers were required to still verify the data in the systems with that in the patient files before submitting reports. For those that were not point of care, data was entered retrospectively from the patient files.

EMRs systems were particularly useful in obtaining HIV-indicators for Care and Treatment (CRT) using queries. Hence, EMRs systems played a role in reporting by contributing to faster data collection, thus reducing the time taken by data officers in collecting data for various

indicators from registers. The ease of reporting attributed by EMRs systems can be described using the following responses from respondents:

> *"It is very easy compared to the registers. If I was to use the registers when doing this work, I would have a lot of calculators and tally sheets. If you see the registers that were there then, there was a lot of paperwork, a lot of time, a lot of documentation. When you have something that actually sums it up and all you have to do is little."*–**Data officer facility I (outlier performer)**

> *"It makes it easy for me to get data because I just run the queries for the data that I want. Instead of going back to the files to count the data one by one how many have been started on treatment, and looking for the ages, those enrolled etc."*–**Data officer facility A (best performer)**

In contrast, facilities where EMRs systems were not implemented had to do the reporting process manually, which was time consuming. Nonetheless, some of the facilities had EMRs system but were not being utilized due to factors such as data migration of files to the EMRs system, lack of human resource designated to use the ERMs system such as clinicians, and system challenges. As stated by the respondents:

> *"Until now they brought in a new clinician, whose now three months in. But if we had this clinician and the system was put in place, it would be running."*–**M&E assistant facility C (best performer)**

> *"It is there but we did not have a clinician. For it to be fully integrated has been a problem. A clinician just came in the other day, now is when she has started to use the EMR system"*–**M&E assistant M&E assistant facility B (best performer)**

Further still, there seemed to be a demarcation regarding acceptability of roles and responsibilities between clinicians hired by the county and those hired by the partners. As one respondent stated:

> *"By the way there is an issue between the clinicians, the staff from the county who are being told to work for a program. So, they will just work. You know a partner needs more than the government when it comes to CCC"*–**M&E assistant facility C (best performer)**

Moreover, in the facilities visited, clinicians interacting with the EMRs system for HIV services were hired by the partners. This enabled clinicians to only focus on CCC as it emerged that clinicians hired by the government had other tasks and were hesitant to perform HIV related tasks, which also included use of EMRs systems in HIV services.

### Reporting timeframe and adherence to reporting deadlines

All the health facilities visited were aware of the reporting deadline and repercussions for late submission of reports. Given that repercussions were imposed for late report submission, facilities were keen on ensuring that both hardcopy and softcopy reports were submitted within the set deadlines regardless of the facility location. The aspect of location came about as an informal observation was made whereby some of the facilities were located inside densely populated slums, which gathered mud during rainy seasons further adding to the challenge of accessibility in and out of the facility.

Health facilities that had no access to DHIS2 were required to submit their reports by 5[th] of every month to their sub-county. On the other hand, deadlines for facilities with access to DHIS2 varied with some facilities having strict deadlines of 5[th], while others having a grace period up to the 10[th] of every month. Nevertheless, time constraint challenges in a bid to beat the reporting deadline were stated among the respondents, as some viewed the reporting period as being too tight. Some of the factors that contributed to this notion include time taken to revise the documentation issues, reporting days falling on weekends, assignment of data officers to more than one reporting site, and parallel reporting.

Documentation issues identified in the reports when conducting data quality checks needed to be corrected before the reports are submitted. In some cases, identifying some of the errors can be time consuming. For example, issues such as identifying "true missing", which results from cases where the reported numbers are too low thus raising an alarm. Hence, finding the correct number of people who missed their appointments can be time consuming and may contribute to delays in reporting. As stated by one respondent who was correcting missed appointments at the time of the visit:

> *"My biggest challenge is timelines. Like you see just know they are asking me to send the corrections, which I have not sent and I am actually working on. So, timelines is the issue."–**Data officer facility I (outlier performer)***

Another issue is reporting dates falling on weekends. This means that the weekend takes up the reporting days, which then reduces the number of days for reporting. This was an issue of concern in most facilities visited, as the deadlines are not extended regardless of the circumstance. As some of the respondents stated:

> *"But there is this time when weekend takes the whole of your date 1, 2, 3 and then date 4 is Monday, that is when you are starting reporting and you are supposed to submit by date 5. Which is tomorrow, that is a challenge"–**Data officer facility K (outlier performer)***

> *"Sometimes when the weekend eats up the reporting time, it's a challenge."–**Data officer facility A (best performer)***

Further still, there are data officers who have been assigned more than one facility to support in HIV-indicator reporting. Given the few reporting days, they are often required to organize their schedules to ensure that reports are submitted on time. In some cases, this may lead to delay as some facilities may be given more priority than others. As stated by one of the respondents:

> Sometimes I may focus on another facility. Because this is a small facility, it may just take 3 hours to finish my report. So by the time I come here, they have submitted the other reports and only remain with the MOH-731. So I will just do it and take it to sub county or give them to submit to sub county. So, I don't know which day it goes. But when it is me who takes it, I normally take it late after the deadline.–**M&E assistant facility G (poor performer)**

> *No. I don't put the same day. I also have another facility which I support, so you find first day of reporting I am here then second day of reporting I am there.–**M&E assistant facility B (best performer)***

Furthermore, most of the facilities are funded hence required to do reports required by the MoH (MOH-731) and those required by supporting partners -Data for Accountability

Transparency and Impact (DATIM). As such, these two HIV summary reports are required per facility in cases where funding is involved, which may be tasking for a data officer reporting for more than one facility. One of the respondents termed this as "roving".

> "... but we have people that rove. But I don't think roving gives someone concentration to ensure quality. Because you have to think of two things at a time. I think it should be one individual per facility."–**Data officer facility L (outlier performer)**

## Access rights and availability of national aggregate reporting system

For facilities to be authorized with credentials to access DHIS2, they have to meet a certain criterion, which the sub-county assesses. As such, not all facilities have access rights to the DHIS2. As such, they are required to submit hard copy MOH-731 reports to the sub-county in order for the reports to be entered in the system. Nevertheless, sub-counties especially those with many constituencies are faced with challenges such as lack of capacity to ensure that all reports are entered on time. This is because, unlike ordinary data entry, MOH-731 reports require numbers to tally otherwise the data will be questionable. DHIS2 has data quality mechanisms that flag these errors within the data upon data entry. This enables facilities to detect and amend questionable data before report submission.

As such, in situations where health facilities with no access-rights to DHIS2 submit hard copy MOH-731 reports for data entry at the sub-countiesMOH-731, errors were more likely to be encountered by the health records information officers (HRIOs) during data entry to DHIS2. As a result, sub-county HRIOs are tasked with contacting facilities through phone calls in order to clarify the errors encountered in data. As stated by one health records information officer at the sub-county:

> "The goodness with the system, for example if you get somebody has put that they have counselled 20 people, and they tested 19, the system flags out, it will alert that why would this facility counsel 20 people and test 19? Where is the 1? So it is now up to me to take my phone and make a call to that facility. You see now they explain. Maybe if it was a transcription error, we are able to go there, because you know they need to correct from the source documents. They don't just tell me "oh no it was wrong, it is 20/ 20". No, we need to go there and counter check, we sign and they sign against that number, so that we are able to put it in the KHIS." - **sub-county Health records Information Officer**

A major issue that emerged during an interview at one of the sub-county offices revealed that the ratio of sub-county officer's to facilities is low in that, the number of facilities are more than sub-county HRIOs. For instance, there are sub-counties with four officers (including volunteers) working with over 50 facilities, which do not submit data to DHIS2 at facility level.

This leads to late data entry to DHIS2, which then affects timeliness. Facilities categorized with outlier performance seemed to have been plagued with this issue based on the interviews conducted. As one respondent stated:

> "... There was a time they (facility) never had the credentials for DHIS2, so I could not upload. So, it depends when did they (sub county) upload."–**Data officer facility I (outlier performer)**

In addition, it is important for systems to be available whenever they are needed for use. Otherwise, this may frustrate efforts of users in trying to accomplish their tasks. Interviewees who used EMRs systems reported that it was always available when needed. This is because the

EMRs systems did not require internet connection. However, there were varied concerns regarding availability of the DHIS2 especially during reporting periods. Given that DHIS2 requires internet connection, some of the issues highlighted include lagging of the internet connection and system down times. As some of the respondents stated:

"…. *When the internet is not working properly and DHIS2 sometimes has an issue, it says bad gateway. So, when DHIS itself has an issue it can delay me.*"–**Data officer facility A (best performer)**

"*Some of these delays are also caused by the system. It goes down, it takes time to load, sometimes we do not have network, and sometimes the system has just gone down*"–**sub-county Health records Information Officer**

"*Then we have internet issues, at times the internet is slow, so inputting the data to dhis2 you have to go an extra mile do it in the evening or very early in the morning when it is not crowded.*"–**Data officer facility J (average performer)**

System availability issues in DHIS2 were also reported at the sub-county office visited. Therefore, absence of system availability contributed to delays in reporting or slowing down the reporting process at both facility and sub-county level.

## Complexity of reports, staff rotations and role of mentorship in reporting

The reports in the MOH-731 were either categorized as complex or easy by respondents, based on the number of registers required, number of indicators, and amount of knowledge required to understand the indicators. CRT reports were among those considered by interviewees as complex, which required knowledge and understanding of the indicators. These reports also took more time to prepare compared to the others especially in facilities with more workload. These reports were also likely to have more documentation issues compared to the others. A HRIO at the sub-county whose office receives reports from various facilities stated as follows:

"*Same to Care and treatment, the indicators are very technical for our health care workers to understand. So you find a lot of transcription errors. It is data that for you to rely on, it needs serious people, people who really understand the data.*"–**sub-county Health records Information Officer**

Documentation issues identified include wrong calculations, gaps within the data, legibility of written figures, misinterpretation of the indicators and codes, and failure to update the registers. Wrong calculations arose in situations where the figures aggregated by nurses or clinicians did not match with those of the data officers. Gaps within the data include scenarios where for instance ten people were counselled and only nine were tested, or ten pregnant women were identified as positive and only five were given prophylaxis. These documentation issues need to be corrected by the data officers before submission of the reports. Documentation issues therefore contribute to delays or slowing down the reporting process more so in cases where there is parallel reporting, which entailed submitting similar reports to both the MoH (submission of MOH-731) and supporting partners (submission of DATIM). As such, data reported in DATIM should be similar to that reported in the MOH-731. A challenge posed by interviewees was amending both reporting tools when documentation issues are encountered. Responses from interviewees on issues of documentation are as follows:

*"...since I had to submit both datims and 731 by 5th, it was hectic. You know you have to work on the negligible stuff like numbers, they don't tally, and then you find an undocumented register, not well documented, it still delays you."*–**Data officer facility I (outlier performer)**

*"... or if something was missed in the register and you want to report, they have to fill the gap and you know for instance a HEI was drawn a PCR at 24 months that is 2 years later, if you do not fill in the results and then you are supposed to report on the outcome or IPT register, anyway the challenge would be I the clinician or nurse is not filling in the register. "*–**Data officer facility K (outlier performer)**

Gaps were also identified in some of the reports perused during interviews, which had gone unidentified in previous years. Post-Exposure Prophylaxis (PEP), Voluntary Medical Male Circumcision (VMMC) and Blood Safety (BS) were among reports considered to be easy as they contained few indicators. In addition, majority of facilities did not offer VMMC and BS, (replaced with Methadone Assisted Therapy at time of study) hence reporting these indicators was easy as it was a matter of just recording zeros. In addition, for facilities that offered VMMC, it happened occasionally especially when students where on holiday as this facilitated performing of the minor circumcision surgeries.

Another contributing factor to documentation is the issue of rotation of nurses involved in reporting and staff turn overs. For instance, at facility level, nurses that were in one service area may be transferred to another service area hence need to be retrained in order to prevent documentation issues. These staff rotations and staff turnovers can be cumbersome to some of the data officers as expressed in the interviewees' responses:

*"...you have completely new people who have never even seen an ANC register, you take them indicator by indicator. Then when they catch up, they leave. Then you start all over again"*–**Data officer facility I (outlier performer)**

*"But the blunder in private facilities is high staff turnover, there is this staff who is doing a good job this month the next month that one got greener pastures, there is a new one again. And then you start again mentoring after a couple of weeks that one again disappears."*–***sub-county Health records Information Officer***

Data officers therefore conduct mentorships through training of indicators and HIV reporting at the various service points, whereas supportive supervisions are conducted on need be basis by the sub-county health records information officers. This in turn promotes better documentation of registers. This is also intended to decrease delays during the reporting period caused by time taken to amend documentation issues.

### Fit between individual, task and technology in reporting

A salient aspect in the reporting process for an individual is the fit between task and fit between technology. This means that the individual has the right competency for the task and technology or tools used to complete the task [26]. All facilities apart from one in this study had employed data officers or M&E assistants, which ensured fit between individual, technology and task. Moreover, all the data officers or M&E assistants interviewed had studied at least health records or both health records and information technology. In addition, those utilizing EMRs systems had received on job training on using various aspects within the EMRs system such as using queries to retrieve data, and trouble shooting. The facility with no data officer happened to be a private facility, and a nurse and a pharmacist were in charge of reporting.

Thus, a misfit between individual and task was identified. As such, this was extra work to the pharmacist who doubled up in doing the reports as stated:

"*You see this is extra work aside from my main work. I told the people form the ministry to give the nurse to be doing this work. They said nurse cannot order drugs, so I said I will just do it because it is once a month.* **Pharmacist facility D (average performer)**

Hence, using the existing employees to multitask in reporting rather than employing a data officer has a potentially negative impact on the data quality of reports as well as motivation of the employees.

## Motivation and awareness of reporting performance

Factors contributing to motivation in reporting among the interviewees include passion for the work, patient well-being, good performance appraisal feedback, gaining insights from the data and support from supervisors. Moreover, it emerged that the presence of motivation implicitly contributes to data quality in terms of accuracy, completeness, and timeliness in reporting. This was revealed by how the respondents reported a sense of commitment towards the patients, which prompts them to strive for good reporting as stated:

"*You know a camera guy who always gives good shots but is never seen in the pictures. I am the guy that I will not see the patient, but I will make sure that I tally everyone. I know everyone by their numbers, and I see their impact in my numbers. Probably I have touched a life somewhere by doing what I did. That motivates me to continue. I made an impact to someone's life when we found out they were positive and I reported. I will give a tally of everyone who came in the facility. It's like you have a role, it's not a main role, you are behind the scenes. That keeps me motivated. It's a good feel.*"–**Data officer facility I (outlier performer)**

"*First, this profession is about people. When I fail to commit to them, I am killing one or two people there. I become motivated when I see them come again. If I do it in a negative way, there will be a lot of murmurs. It is out of passion as well.*"–**Data officer facility H (poor performer)**

Awareness of facility reporting performance also provides knowledge as to whether a facility is meeting reporting requirements for completeness and timeliness. Moreover, good reporting performance not only portrays a facility in good light but also for the data officer(s) in charge during a particular reporting period. Nonetheless, performance reports for facility completeness and timelines in DHIS2 were not utilized by respondents for purposes of identifying respective facility reporting performance. As such, as long as the report was submitted within the submission deadline and without questioning of the data, facility reporting completion and timeliness requirement was considered having being met. One of the responses on this is as stated:

"*As long as I have sent it be 5th in terms of timeliness, I know I have performed well. And if they have not kept on calling me all the time to ask me about the data, then I know I have done well.*"–**Data officer facility B (best performer)**

As such, data officers often relied on the approaches used by various respective health facilities key administrators to convey feedback on facility and individual reporting performance. Some of the feedback channels include through WhatsApp groups, through in-charges, during

data review meetings, and during performance appraisals. One of the responses on this by an interviewee is as stated:

> "*Yes we always have a data review whereby you see your performance, they also project the way you keep reporting. If you are in green, it means your reports are always on time, if you are on yellow you try but sometimes you are late, if you are red, you are always late. But no major repercussion, just try to not submit the reports late or pressure will mount on you.*" **Data officer facility K (outlier performer)**

Hence, awareness of performance also implicitly contributed to motivation in reporting especially when feedback is good. Another factor that could potentially contribute to motivation is increasing space in records offices in some of the facilities. A general informal observation revealed some of the office spaces to be quite small in some of the facilities with some of the respondents eluding that more space would be good for them.

## Availability of standard operating procedures, training, and supervision

Among the facilities visited, those funded had existing Standard Operating Procedures (SOPs) that were developed by the supporting partners. An informal observation was made whereby some of the facilities had put copies of their SOPs on the walls, while others had theirs only stored in cabinets. The SOPs regarding data quality were examined in order to understand the procedures put in place. The SOPs for data quality laid down roles and responsibilities for data management, data quality assurance hence providing guidance to data officers. Furthermore, internal monthly data review meetings were carried out in these facilities in order to review the HIV indicator data quality. These meeting mostly comprised of nurses, clinicians, and data officer employed by the partners.

Training is a very salient component in reporting. Nonetheless it was provided mostly by supporting partners for their employees. Training of the MOH-731 tools were provided only once when the tools had been updated. Nonetheless, supportive supervisions from sub-county doubled up as training in some facilities. Some facilities reported having had sub-county supervisions at least once a year, while others were hesitant in admitting to having no visits from the sub-county. Some of the responses are as follows:

> "*For the county no, but for the program if they organize, they call us.*"–**Data officer facility L (outlier performer)**

> "*That is the challenge. We rarely get trainings. We lack good training, just good training. We go for some data quality assessment after three months but that is not enough. Because things to do with information, they transform and keep changing.*"–**Data officer facility H (poor performer)**

Another emerging issue on training was the dependability on availability of funding in order for trainings to take place as stated by some of the respondents:

> *Yea, now that the is no funding, that (training) is once in our dreams.*–**Data officer facility B (average performer)**

> *It also depends on funding. When there was money we used to go monthly, then it changed to quarterly. That is training for all of us because these tools keep on changing. There is a time we went 1 week for training because of these new tools.*–**Data officer facility A (best performer)**

Hence, factors contributing to frequency of training and supervision were attributed to availability of funding, and availability of sufficient human resources to conduct the trainings, especially supportive supervision, which relied on sub-county staff.

## Discussion

This qualitative case study identified barriers and facilitators in HIV-indicator reporting that were linked to the RHIS reporting process and determinants of RHIS (technical, behavioral and organizational determinants). Facilities performing well and those performing poorly might have contextual differences that affect their performance, hence influencing the barriers or facilitators faced as posited in other studies [27,28].

This study revealed that whereas facilities may demonstrate differences in reporting performance, they are likely to face similar barriers and facilitators regardless of the contextual differences. For instance, it was assumed that facilities with an EMRs system were likely to have good reporting performance. This is because these facilities are required to have availability of trained personnel, appropriate infrastructure, adequate security, support and maintenance protocols, and accessible management support prior to EMRs system implementation [29]. Nonetheless, it emerged that all facilities that performed poorly in timeliness (outlier performance group) had EMRs system implementations. In addition, among the facilities that had high reporting performance in completeness and timelines (best performing group), only one had a functioning retrospective stand-alone EMRs system, whereas the rest had EMRs systems that were not in use due to lack of human resource. Therefore, EMRs systems implementation in facilities did not translate to good performance in reporting to DHIS2.

This can also be attributed to the lack of interoperability between the EMRs systems and DHIS2 to enable seamless data transmission. This lack of interoperability between EMRs systems and national aggregate systems remains a challenge in LMICs, leading to systems operating in silos [29,30]. As such, given that EMRs systems have been attributed as having the potential to dramatically reduce the data collection burden by automating the reporting process [30,31], this potential has not yet been realized as revealed in our findings. Nonetheless, utilization of EMRs system in reporting still contributed to easing the data collection burden, as data was retrieved from EMRs systems rather than multiple registers, which is time consuming. Furthermore, facilities that utilized EMRs systems reported to have obtained technical support and training when required through their supporting partners. This is an indication of a step towards the right direction, as compared to previous studies, which indicate challenges such as lack of IT support [11,32].

A number of issues were also identified, which contributed to facilities performing poorly in timeliness. These include issues such as, time-consuming efforts to correct data quality issues in reports, and submission of reports by hand to the sub-county office. As such, insufficient human resources, lack of availability of the DHIS2 during reporting, and slow internet connection at the sub-county level resulted to late entries of the reports submitted by hand, which then resulted to poor performance in timeliness among facilities. This then hinders the repercussion measures that had been put in place to deter late reporting. This is because reports maybe submitted on time to the sub-counties by facilities in order to avoid repercussions for late reporting, however, they are entered late in the system at the sub-county level.

Nonetheless, efforts have been made by sub-counties to offload the data entry burden by ensuring facilities that meet a predefined criterion have access rights to DHIS2. As such, despite some of the facilities having access to DHIS2 at the time of study, it emerged that this was not the case in previous years. Hence, it was assumed that the lack of access rights to DHIS2 might have also been a contributor to poor performance in timeliness.

Supportive supervisions and mentorship have been identified as good approaches that contribute to providing quality data for M&E [11,33]. Nonetheless, insufficient human resource at the sub-county limits frequent supportive supervision at the facility level. Moreover, although on job training was provided in facilities funded by supporting partners, facilities that did not receive external funding depended on trainings and supportive supervision provided by sub-county. Nonetheless, supervision was not conducted as frequently as expected based on respondents' perspectives. Only two facilities among those assessed were not funded by supporting partners and therefore lacked frequent on job training. In addition, staff rotations and staff transfer frustrated supportive supervision and mentorship efforts in the sub-county, both in facilities with external funding, and those without. This was because they brought about demand to retrain incoming staff, which proved to be tasking as reiterated by respondents.

Our findings also revealed time constraints echoed by facilities in all performance groups as a major concern during the reporting period. These time constraints were further aggravated by issues such as, reporting period falling on a weekend (which meant that a day or two were deducted from the reporting period) and data officers being assigned to more than one health facility, which led to some facilities being given less priority compared to others in reporting MOH-731. As such, dealing with more than one facility has the potential to heighten the chances of reporting burden and risk of hampering data quality. In addition findings from this study reveal parallel reporting of HIV-indicator by facilities funded by supporting partners, which continues to be a challenge facing M&E systems in LMICs [11,34,35]. Time constraints issues are further aggravated by documentation errors brought about by staff rotation and staff transfer of health workers involved in reporting, lack of understanding the indicators, and DHIS2 availability issues, which further slowed down the reporting process. Ledikwe et al. also reported similar issues related on data gaps as result of changes in staffing [11].

In order to ensure timeliness in reporting, a resource intensive approach would entail strengthening capacity in health facilities through rigorous staff training, strong internet connection in facilities with computers, ensuring EMRs systems lying dormant in facilities are utilized for reporting and ensuring many facilities are able to meet requirements needed to obtain access rights to DHIS2 in order to perform data entry for themselves. These will facilitate timely submission of good quality reports to DHIS2. Nonetheless, a less resource intensive approach will entail ensuring availability of sufficient skilled human resources at the sub-county level to perform data entry tasks and provide supportive supervision at facility level. Continuous training of health workers on reporting will also enhance data quality and fill the gap left through rotations and staff turnovers.

This study has identified facilitators and barriers in HIV reporting among facilities with various performances in completeness and timeliness in reporting HIV-indicators to DHIS2. Findings reveal that similar barriers and facilitators are shared across the different performances. Nonetheless, it emerged that skilled human resources involved in reporting, in combination with access to DHIS2 promote better performance of facilities reporting completeness and timeliness.

A limitation of this assessment was that there had been staff turnovers and rotations among data officers in some of the facilities, which made it difficult to provide conclusive descriptions of happenings of previous years. In addition, though we used cases in one county, we expect that the findings revealed in this assessment are transferrable in other counties and in LMICs in similar contexts. In addition, given that facility performance was not based on indicator data completeness and accuracy, future verification exercises are warranted for the various facility performance categories. Further still, there have been efforts by the MoH in availing training tools such as eLearning portals that provide courses for DHIS2 and HIV M&E, which can provide training for facility health workers. Nonetheless, we did not delve into usage of

such portals, hence future studies can be conducted in order to assess the benefit of these courses in relation to performance in data quality.

## Conclusion

The study identified barriers and facilitators linked to the RHIS process and determinants of RHIS, which include three interrelated factors, technical, organizational and behavioral. The findings demonstrated that whereas facilities may demonstrate different performances in completeness and timeliness in reporting, the barriers and facilitators that they face may be less different among them. It was expected that EMRs systems would improve reporting, and in order to realize their potential in reporting, there needs to be integration of DHIS2 and EMRs systems as posited in feasibility studies conducted [31]. This study could not attribute best performers to presence of EMRs systems. Nonetheless, future prospects to automate indicator reporting between EMRs systems and DHIS2 will pave way to determine whether best performing facilities is accelerated by use of EMRs system. Continuous evaluations have been advocated within health information systems literature. Therefore, continuous qualitative assessments are also necessary in order to determine improvements, as well as recurring of similar issues based on previous assessments. These assessments have also complemented other quantitative analyses related to this study [21].

## Supporting information

**S1 Appendix. Interview guide.**
(DOCX)

## Acknowledgments

We would like to thank all the respondents who took part in this qualitative analysis.

## Author Contributions

**Conceptualization:** Milka B. Gesicho, Ankica Babic.

**Formal analysis:** Milka B. Gesicho.

**Investigation:** Milka B. Gesicho.

**Methodology:** Milka B. Gesicho, Ankica Babic.

**Supervision:** Ankica Babic.

**Validation:** Ankica Babic.

**Writing – original draft:** Milka B. Gesicho.

**Writing – review & editing:** Milka B. Gesicho.

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
