## [Decision Letter · Decision Letter 0]

20 Jan 2021

PONE-D-20-39745

Identifying barriers and facilitators in HIV-indicator reporting for different health facility performances: A qualitative case study

PLOS ONE

Dear Dr. Gesicho,

Thank you for submitting your manuscript to PLOS ONE. After careful consideration, we feel that it has merit but does not fully meet PLOS ONE’s publication criteria as it currently stands. Therefore, we invite you to submit a revised version of the manuscript that addresses the points raised during the review process.

We look forward to receiving your revised manuscript.

Kind regards,

Vijayaprasad Gopichandran

Academic Editor

PLOS ONE

Journal Requirements:

Reviewers' comments:

Reviewer's Responses to Questions

**Comments to the Author**

1. Is the manuscript technically sound, and do the data support the conclusions?

Reviewer #1: Yes

Reviewer #2: Yes

2. Has the statistical analysis been performed appropriately and rigorously? 

Reviewer #1: N/A

Reviewer #2: N/A

3. Have the authors made all data underlying the findings in their manuscript fully available?

Reviewer #1: Yes

Reviewer #2: Yes

4. Is the manuscript presented in an intelligible fashion and written in standard English?

Reviewer #1: Yes

Reviewer #2: Yes

5. Review Comments to the Author

Reviewer #1: Overall Comments:

Thank you for giving me the opportunity to review such an interesting manuscript. The importance of this manuscript is clear to me. However, there were several grammatical and punctual errors throughout the text which must be addressed in order to improve the readability of this work. Additionally, key information is lacking from the methods section which are needed to ensure the rigour of this work.

Abstract

•Line 33-34 “nonetheless, lack of …” seems to be repetitive (it was already mentioned in line 30).

•It would be helpful to note where you are focusing your efforts (i.e., indicate that the 13 facilities you are working with are located in Kenya).

Introduction

•Line 62-63 needs revision. Are you implying that data is entered directly to DHIS2 and that paper-based summary forms are not often used to transfer data? Or are you saying that these data are seldom collected in general?

Methods

•What constitutes “best performers, average performers, poor performers, and outlier performers?” I.e., what percentage of timeliness and completeness did facilities have to meet in order to receive these labels?

•How exactly did you choose which facilities you wanted to include in your case?

•Did you choose your facilities based on a maximum variation sampling method (i.e., 3 cases of best performers, 3 of average, 3 of poor, and 3 of outlier?)? Or was a purposive sampling strategy used? If purposive, why were certain groups chosen?

•Line 117-118, purposive sampling was used to identify key informants – what indicator was used to constitute an individual as a key informant? Are you saying that those who were in charge of reporting (line 119) are the key informants? If so, this must be made clearer.

•How were interview data analyzed? Were one or two individuals responsible for analysis? If one person was involved, were strategies taken to reduce bias in interpretation? If two people were involved, were techniques used to ensure strong interrater reliability?

Results

•Although I found the results to be incredibly interesting, their presentation can be considerably condensed. One or two respondent quotations per section could potentially be sufficient in getting the point across.

Reviewer #2: Thanks for the invitation to review this manuscript.

Introduction: This article provides a useful summary of barriers and facilitators in HIV-indicator reporting. The authors make a critical contribution to knowledge, especially now when most LMICs strive to improve data to guide decisions and health systems’ improvement. The need for the study is elaborated. However, a few minor observations need clarification:

1.In the data collection section (line 125 – 126), it is interesting that one of the methods employed to collect data for this manuscript was the informal direct observation. However, it is not clear what data was collected using this method as it is not reported in this manuscript

2.In the ethics section, the article states that privacy and confidentiality were ensured. However, it is not clear as to how was this done?

3.The reporting of quotes needs consistence. While some quotes indicate the interviewer number, others are silent. For example, line 459

4.The manuscript may need some general formatting, especially deleting empty lines in some places between paragraphs and quotes – see line 442, 532

5.Drawing from the results, especially that there are differences between county and partner employees in terms of acceptability of roles, how can LMICs build sustainable health information systems in the absence of partners?

6.The paper would also benefit from a thorough and close editorial review of grammar. For example,

-line 207 – is an incomplete phrase

-line 275 – change “contributed” to contributing

-line 295 – edit – facilities were EMRs where not implemented

-line 320 – edit – where aware of the reporting deadline

6. PLOS authors have the option to publish the peer review history of their article (what does this mean?). If published, this will include your full peer review and any attached files.

Reviewer #1: **Yes: **Anish Arora

Reviewer #2: **Yes: **Daniel Nyato

---

## [Author Response · Author response to Decision Letter 0]

6 Feb 2021

Dear Scientific Program Committee,

Re: Manuscript PONE-D-20-39745: Identifying barriers and facilitators in HIV-indicator reporting for different health facility performances: A qualitative case study

We appreciate the review by the PLOS One Scientific Program Committee of our Manuscript titled “Identifying barriers and facilitators in HIV-indicator reporting for different health facility performances: A qualitative case study". We also appreciate the opportunity to respond comprehensively to the reviewers’ comments.

Please find our responses to all the comments by the reviewers below:

Reviewer #1: Overall Comments:

Thank you for giving me the opportunity to review such an interesting manuscript. The importance of this manuscript is clear to me. However, there were several grammatical and punctual errors throughout the text which must be addressed in order to improve the readability of this work. Additionally, key information is lacking from the methods section which are needed to ensure the rigour of this work.

Abstract

•Line 33-34 “nonetheless, lack of …” seems to be repetitive (it was already mentioned in line 30).

We appreciate and agree with the reviewer’s comments. We have removed the repetitive sentence. 

•It would be helpful to note where you are focusing your efforts (i.e., indicate that the 13 facilities you are working with are located in Kenya).

We appreciate and agree with the reviewer’s comments and have indicated the location of the facilities in the manuscript with track changes (line 25-27) as outlines below: '

“Data was collected using semi-structured in-depth interviews with 13 participants, and included archival records, look into documentation, and informal direct observation at 13 facilities in Kenya.”

Introduction

•Line 62-63 needs revision. Are you implying that data is entered directly to DHIS2 and that paper-based summary forms are not often used to transfer data? Or are you saying that these data are seldom collected in general?

Thank you for this observation. What was meant here is that facilities use paper-based summary forms to collect aggregate data. This data is then entered directly to DHIS2 at facility level for those with DHIS2 access rights. Facilities with no access rights submit the reports to the subcounty to be entered directly to DHIS2. The term ‘seldom’ was erroneous in this sentence and has been revised in the manuscript with track changes (line 62-63) as follows: 

“Therefore, aggregate indicator data from various HIV services are collected using paper-based summary forms at the facility level, which are then entered into the DHIS2.”

Methods

•What constitutes “best performers, average performers, poor performers, and outlier performers?” I.e., what percentage of timeliness and completeness did facilities have to meet in order to receive these labels?

Thank you for this question. The categorization of the four performance groups was based on a cluster analysis using k-means clustering algorithm, which was used to group facilities based on performance in reporting completeness and timeliness. The results for this analysis are discussed in detail in our two separate published papers, which have been cited in the manuscript to provide further details. 

Below is a section extracted from the cited papers, with details of the performance groups. 

The four clusters were characterized based on health facility performance as follows:

Best performers: This cluster consisted of health facilities that had the highest percentage in reporting completeness and timeliness in a particular reporting year. 

Average performers: This cluster consisted of health facilities that had lower percentage in reporting completeness and timeliness compared to best performers in a particular year. 

Poor performers: This cluster consisted of health facilities with lowest percentage in reporting completeness and timeliness in a particular year. 

Outlier performers: This cluster consisted of health facilities with high percentage in completeness compared to average performers, but with low percentage in timeliness in that particular year. 

•How exactly did you choose which facilities you wanted to include in your case?

Purposive sampling was used to select the facilities based on the criteria mentioned in line (101-103).

“healthcare facilities that meet the following criteria (i) located in Nairobi (ii) either use EMRs system or paper in reporting, (iii) reporting performance indicators (facility reporting completeness and timeliness).”

•Did you choose your facilities based on a maximum variation sampling method (i.e., 3 cases of best performers, 3 of average, 3 of poor, and 3 of outlier?)? Or was a purposive sampling strategy used? If purposive, why were certain groups chosen?

The type of Purposive sampling used to select facilities in the four performance groups was stratified purposeful sampling and this was based on reporting performance. There were two main categories included, health facilities performing well (best=3, average=3,) and health facilities performing poorly (outlier=4, poor=2). This detail has been included in the manuscript with track changes line 122-128. 

“Hence, the type of purposive sampling used was stratified purposeful sampling, whereby health facilities from the two levels were selected based on reporting performance.” Line 106-108

“In this study, performance was categorized in two main groups, facilities performing well (best performers = 3, average performers =3) and facilities performing poorly (outlier performers= 4 and poor performers=2).” Line 123

•Line 117-118, purposive sampling was used to identify key informants – what indicator was used to constitute an individual as a key informant? Are you saying that those who were in charge of reporting (line 119) are the key informants? If so, this must be made clearer.

Thank you for this response. We have revised the sentences to make it clearer that those in charge of reporting are key informants (line 130-132) as outlined below: 

“Therefore, the key informants who in this study are the units of analysis included personnel in charge of reporting as they serve as the focal point around which all reporting activities take place.”

•How were interview data analyzed? Were one or two individuals responsible for analysis? If one person was involved, were strategies taken to reduce bias in interpretation? If two people were involved, were techniques used to ensure strong interrater reliability?

The interview data was analyzed by one individual. In order to reduce bias, peer debriefing advocated by Creswell (cited in manuscript) was conducted with the second author. This involved having someone familiar with the research asking hard questions on methods and interpretations.

“Peer debriefing was also carried out in sessions after conducting two to three interviews and during analyses” line 200-201

Results

•Although I found the results to be incredibly interesting, their presentation can be considerably condensed. One or two respondent quotations per section could potentially be sufficient in getting the point across.

We appreciate the reviewers comment on this. We have condensed the respondent quotations where possible per section as advised, thus having one or two quotes supporting a finding. 

Reviewer #2: Thanks for the invitation to review this manuscript.

Introduction: This article provides a useful summary of barriers and facilitators in HIV-indicator

reporting. The authors make a critical contribution to knowledge, especially now when most

LMICs strive to improve data to guide decisions and health systems’ improvement. The need for the study is elaborated. However, a few minor observations need clarification:

1. In the data collection section (line 125 – 126), it is interesting that one of the methods

employed to collect data for this manuscript was the informal direct observation.

However, it is not clear what data was collected using this method as it is not reported in

this manuscript

We appreciate your observations. We have included more sentences to indicate the data collected using informal observations.

Informal observation was used in observing the following areas as outlined in the manuscript with track changes 

Line 353-356

“The aspect of location came about as an observation was made whereby some of the facilities were located inside densely populated slums, which gathered mud during rainy seasons further adding to the challenge of accessibility in and out of the facility.”

Line 229-233

“An informal observation made when conducting interviews was the interaction between the nurses, clinicians and the data officer, which revealed a sense of putting in effort to provide the data required by the data officers. For instance, in some facilities, an observation was made on nurses bringing updates on aggregated data written on paper to the data officer”

Line 624-625

“An informal observation was made whereby some of the facilities had put the SOPs on the walls, while others had them stored in cabinets.”

2. In the ethics section, the article states that privacy and confidentiality were ensured.

However, it is not clear as to how was this done?

We appreciate the reviewers comment on this. We have revised the sentences to describe how privacy and confidentiality was ensured. Ethical considerations section Line 175-176

“Privacy and confidentiality were ensured by not revealing the identities of the participants nor the facilities that took part in this study”.

3. The reporting of quotes needs consistence. While some quotes indicate the interviewer

number, others are silent. For example, line 459

Thank you for the comments. We have revised the quotes by indicating and highlighting the respondents details at the end of the quotation and spacing between quotes in order to ensure consistency as outlined in the example below from the manuscript.

“For the county no, but for the program if they organize, they call us.” – Data officer facility L (outlier performer) 

“That is the challenge. We rarely get trainings. We lack good training, just good training. We go for some data quality assessment after three months but that is not enough. Because things to do with information, they transform and keep changing.” – Data officer facility H (average performer) 

4. The manuscript may need some general formatting, especially deleting empty lines in

some places between paragraphs and quotes – see line 442, 532

Thank you for the comment. We have re-read and formatted the manuscript appropriately as advised. 

5. Drawing from the results, especially that there are differences between county and

partner employees in terms of acceptability of roles, how can LMICs build sustainable

health information systems in the absence of partners?

We appreciate the reviewer’s comments. In our opinion LMICs can build sustainable HIS in absence of partners by filling in the financial gaps left by funding agencies through allocation of more national finances towards HIV treatment and investing in staff training and staff retention. 

6. The paper would also benefit from a thorough and close editorial review of grammar. For

example,

- line 207 – is an incomplete phrase

- line 275 – change “contributed” to contributing

- line 295 – edit – facilities were EMRs where not implemented

- line 320 – edit – where aware of the reporting deadline

We appreciate and agree with the reviewer’s comment. We have reviewed the paper for grammar and revisions have been made appropriately.

---

## [Editor Report · Decision Letter 1]

9 Feb 2021

Identifying barriers and facilitators in HIV-indicator reporting for different health facility performances: A qualitative case study

PONE-D-20-39745R1

Dear Dr. Gesicho,

We’re pleased to inform you that your manuscript has been judged scientifically suitable for publication and will be formally accepted for publication once it meets all outstanding technical requirements.

Kind regards,

Vijayaprasad Gopichandran

Academic Editor

PLOS ONE
---

## [Editor Report · Acceptance letter]

12 Feb 2021

PONE-D-20-39745R1 

Identifying barriers and facilitators in HIV-indicator reporting for different health facility performances: A qualitative case study 

Dear Dr. Gesicho:

I'm pleased to inform you that your manuscript has been deemed suitable for publication in PLOS ONE. Congratulations! Your manuscript is now with our production department. 

Kind regards, 

on behalf of

Dr. Vijayaprasad Gopichandran 

Academic Editor

PLOS ONE